# Unraveling the Complexities of Toll-like Receptors: From Molecular Mechanisms to Clinical Applications

**DOI:** 10.3390/ijms25095037

**Published:** 2024-05-05

**Authors:** Yi-Hsin Chen, Kang-Hsi Wu, Han-Ping Wu

**Affiliations:** 1Department of Nephrology, Taichung Tzu Chi Hospital, Taichung 427, Taiwan; yishin0819@gmail.com; 2School of Medicine, Tzu Chi University, Hualien 97004, Taiwan; 3Department of Artificial Intelligence and Data Science, National Chung Hsing University, Taichung 40227, Taiwan; 4Department of Pediatrics, Chung Shan Medical University Hospital, Taichung 402, Taiwan; 5School of Medicine, Chung Shan Medical University, Taichung 402, Taiwan; 6College of Medicine, Chang Gung University, Taoyuan 33302, Taiwan; 7Department of Pediatrics, Chiayi Chang Gung Memorial Hospital, Chiayi 613016, Taiwan

**Keywords:** Toll-like receptors, immunity, adaptive immunity, innate immunity, ligands

## Abstract

Toll-like receptors (TLRs) are vital components of the innate immune system, serving as the first line of defense against pathogens by recognizing a wide array of molecular patterns. This review summarizes the critical roles of TLRs in immune surveillance and disease pathogenesis, focusing on their structure, signaling pathways, and implications in various disorders. We discuss the molecular intricacies of TLRs, including their ligand specificity, signaling cascades, and the functional consequences of their activation. The involvement of TLRs in infectious diseases, autoimmunity, chronic inflammation, and cancer is explored, highlighting their potential as therapeutic targets. We also examine recent advancements in TLR research, such as the development of specific agonists and antagonists, and their application in immunotherapy and vaccine development. Furthermore, we address the challenges and controversies surrounding TLR research and outline future directions, including the integration of computational modeling and personalized medicine approaches. In conclusion, TLRs represent a promising frontier in medical research, with the potential to significantly impact the development of novel therapeutic strategies for a wide range of diseases.

## 1. Introduction

### 1.1. Overview of Innate Immunity and Pattern Recognition Receptors (PRRs)

Toll-like receptors (TLRs) are integral to the innate immune system as they recognize diverse molecular patterns that trigger immune responses. Humans possess ten functional TLRs (TLR1–10), each specializing in detecting different signals and pathogen-associated molecular patterns, including damage-associated molecular patterns (DAMPs), microbial-associated molecular patterns (MAMPs), pathogen-associated molecular patterns (PAMPs), and xenobiotic-associated molecular patterns (XAMPs). These receptors share a common structural organization, comprising an N-terminal domain with leucine-rich repeats (LRRs), a single transmembrane helix (TM), and a C-terminal cytoplasmic Toll-interleukin-1 receptor (TIR) domain [1]. The aim of this review is to delineate the multifaceted roles of TLRs within the scope of the innate immune system, elucidating their structural characteristics, signaling pathways, and implications in various diseases. This review also aims to discuss recent advancements in TLR research that have clinical relevance, particularly focusing on their therapeutic potential.

In addition to TLRs, the innate immune system is equipped with a robust array of pattern recognition receptors that ensure a comprehensive and adaptable response to pathogens. These include RIG-I-like receptors, which are cytoplasmic detectors of viral RNA, initiating crucial antiviral defenses by producing type I interferons [2]. C-type lectin receptors play a pivotal role in recognizing fungal carbohydrates and bacterial elements, enhancing phagocytosis and mediating direct antimicrobial actions [3]. NOD-like receptors within the cell cytoplasm activate upon sensing microbial motifs or cellular stress signals, triggering inflammasomes that promote cytokine production and drive inflammatory responses [3]. Furthermore, the cGAS-STING pathway marks a critical defense mechanism against cytosolic DNA from viruses or damaged host cells, orchestrating a potent immune activation that includes interferon responses [4].

Studies have elucidated the critical and multifaceted roles of TLRs in the innate immune system. Notably, TLR9 and the cyclic GMP-AMP synthase-stimulator of interferon genes (cGAS-STING) pathway possess DNA-sensing capabilities and play roles in initiating innate immune responses, underscoring the complexity and adaptability of TLR-mediated signaling in response to diverse stimuli [5]. The significance of TLRs in viral infections is further highlighted by reports that TLR3, TLR7, and TLR8, along with cytoplasmic retinoic acid-inducible gene I (RIG-I)-like receptors, are instrumental in initiating innate immune responses against viral nucleic acids [6]. Additionally, research on TLR signaling pathways has emphasized their importance in disease progression such as in herpes simplex virus 1 (HSV-1) infection wherein TLRs and their signaling pathways in the trigeminal ganglia play crucial roles in controlling the progression to encephalitis [7]. This underscores the vital function of TLRs in antiviral defense and their potential as therapeutic targets.

TLRs have been targeted in numerous clinical trials due to their key role in immune system activation and disease pathogenesis, particularly in infectious diseases, cancer, and inflammatory conditions. Clinical trials have systematically explored the use of TLR agonists and antagonists, grouping them based on the targeted pathology and therapy combinations. As illustrated in Table 1, the range of TLRs’ applications is extensive, encompassing not only agonists and antagonists but also their use as vaccine adjuvants, which are critical in infectious disease contexts. For instance, TLR7 and TLR9 are often activated in therapies against viral infections and as adjuvants in cancer vaccines, while TLR2 and TLR4 are targeted by antagonists to mitigate inflammatory responses in sepsis and other inflammatory diseases [8]. Additionally, advancements in understanding TLR-mediated signaling, such as its regulation by ubiquitination and microRNAs, suggest further therapeutic possibilities for targeted manipulation in clinical settings [9].

### 1.2. Brief Introduction to TLRs

TLRs are integral components of the immune system that detect PAMPs and DAMPs. Their activation activates signaling pathways that culminate in inflammatory and immune responses. TLRs are ubiquitously present across various cell types and tissues and play a pivotal role in immune surveillance [6,15]. Upon ligand binding, TLRs initiate a cascade of events that lead to the production of pro-inflammatory cytokines and type I interferons, which are essential for shaping the immune response. Regulation of TLR signaling is crucial for maintaining immune homeostasis; dysregulation is linked to several diseases, including autoimmunity, chronic inflammation, and cancer, positioning TLRs as prime therapeutic targets [6,16]. The significance of TLRs extends to the pathogenesis of lifestyle-related diseases wherein TLR9 recognizes DNA fragments, highlighting the intricate cross-regulation of type I interferon signaling pathways [17]. This is corroborated by studies illustrating the association of endosomal transmembrane TLRs (such as TLR3) with various diseases that can activate the interferon signaling pathway [6]. Furthermore, the role of TLRs in diabetic encephalopathy underlines their impact on signaling pathways by interacting with receptors such as RAGE/Toll-like receptors, which are implicated in several conditions, including cancer, cardiovascular disease, and sepsis [18].

Therapeutic modulation of TLR signaling holds promise for the treatment of a range of diseases. For instance, LAMP3-induced ectopic TLR7 expression in salivary gland epithelial cells and amplification of type I IFN production highlight potential therapeutic strategies for Sjögren’s disease [19]. Moreover, compounds such as piperine, which attenuates hepatic ischemia/reperfusion injury by suppressing the TLR4 signaling cascade, point towards safer options for the clinical treatment and prevention of ischemia-related diseases [20]. In conclusion, the critical role of TLRs in immune surveillance and their involvement in various diseases underscores the importance of proper regulation of TLR signaling. Their potential as therapeutic targets offers avenues for the treatment of autoimmune, inflammatory, and chronic diseases, emphasizing the need for further research in this domain.

### 1.3. Structure and Characterization of TLRs

#### 1.3.1. Structural Features of TLRs

TLRs are type I transmembrane proteins with distinctive structures that play critical roles in the innate immune system. They are known for their extracellular leucine-rich repeats (LRRs), which range from 20 to 27 per TLR, enabling the detection of PAMPs and DAMPs [21]. Recent findings highlight the significant impact of gene polymorphisms within TLRs on their function. For example, polymorphisms in TLR4 have been shown to influence milk production traits in livestock by altering immune responsiveness [22]. Additionally, specific TLR gene polymorphisms are associated with differential susceptibility to autoimmune diseases, further underscoring the importance of considering genetic variability in TLR function [23]. These LRRs form horseshoe-like structures that recognize a variety of molecular signatures of microbial intruders or indicators of cell damage. A pivotal component of TLRs is the Toll/interleukin-1 receptor (TIR) domain, which is located within the intracellular region and is crucial for signal transduction that leads to the activation of genes that drive inflammatory responses [24,25]. Furthermore, the extracellular domains of TLRs are modified by glycan moieties, which are presumed to play a role in ligand binding; however, the exact functions of these modifications remain to be elucidated [21,26].

#### 1.3.2. Differences between TLR Family Members

TLRs are a class of proteins instrumental in the immune system’s first line of defense. Each TLR recognizes specific PAMPs or DAMPs because of their LRR structures, which confer ligand specificity. This specificity enables the immune system to differentiate between various pathogens and damaged host cells, thereby initiating the appropriate immune responses [21]. The TLR family is functionally diverse, with members located on the cell membrane or within intracellular compartments such as the endoplasmic reticulum, endosomes, and lysosomes. Cell membrane TLRs, such as TLR2 in heterodimeric association with TLR1 or TLR6, TLR4, TLR5, and TLR10, primarily respond to microbial membrane components such as lipoproteins, flagellin, and lipopolysaccharides. In contrast, intracellular TLRs, including TLR3, TLR7, TLR8, and TLR9, are specialized in sensing nucleic acids, highlighting the system’s adaptability to various types of molecular signatures indicative of infection or cellular damage [21]. Humans possess ten functional TLRs (TLR1–TLR10), each tailored to detect specific pathogen-associated and damage-associated molecular patterns, reflecting their role in immune surveillance and response. These receptors share a common structural organization comprising an N-terminal domain with leucine-rich repeats (LRRs), a single transmembrane helix (TM), and a C-terminal cytoplasmic Toll-interleukin-1 receptor (TIR) domain [27]. Expression mapping shows widespread but distinct patterns, with variations depending on cell type and tissue. For example, TLR4 is broadly expressed across many cell types and tissues, whereas TLR9 is more restricted to immune cell populations within endosomal compartments, illustrating the functional specialization of TLR locations. Notably, TLR11, TLR12, and TLR13, though not present in humans, are expressed in mice and other species, playing crucial roles in immune responses to various pathogens such as protozoans and bacteria [28]. These TLRs are primarily involved in recognizing specific bacterial components and are essential in initiating appropriate immune responses in these species.

TLR signaling pathways diverge to initiate tailored immune responses. For instance, TLR4 recognition of lipopolysaccharides (LPSs) from Gram-negative bacteria triggers signaling cascades involving adapters such as MyD88 and TRIF, leading to the production of pro-inflammatory cytokines and type I interferons [21]. These mechanisms underscore the role of TLRs in bridging innate and adaptive immunity and ensuring coherent and effective immune defense. The subcellular localization of TLRs plays a crucial role in their function. Cell surface TLRs detect extracellular threats, whereas intracellular TLRs monitor the interior of cells for viral and bacterial nucleic acids, demonstrating a sophisticated surveillance system that protects against a wide array of pathogens [29]. Table 2 demonstrates the primary locations, principal ligands, signaling pathways, and functions of each TLR, offering a detailed comparison of their roles in immune surveillance and disease pathogenesis.

### 1.4. Ligands and Signaling of TLRs

#### 1.4.1. Exogenous Ligands (PAMPs) 

PAMPs include a variety of microbial molecular structures that trigger immune responses, such as flagellin, lipoteichoic acid (LTA), and peptidoglycan (PGN) from Gram-positive bacteria and LPS from Gram-negative bacteria. These components play significant roles in inducing various immune and inflammatory responses, as explored in a study that investigated the synergistic effects of PGN, LTA, and LPS on bovine mammary epithelial cells, revealing insights into transcriptome changes, inflammatory responses, and associated epigenetic mechanisms [34].

#### 1.4.2. Endogenous Ligands (DAMPs)

Endogenous ligands, also known as DAMPs, play a crucial role in immune system activation in response to non-physiological cell death and injury. These molecules, including extracellular matrix components such as hyaluronan and fibrinogen, plasma membrane constituents, nuclear and cytosolic proteins such as high-mobility group box protein 1 (HMGB1) and heat shock proteins, and elements from damaged or fragmented organelles such as mitochondrial DNA. Importantly, galectins, particularly Galectin-3, have been recognized as critical mediators in inflammation by binding to TLRs, notably TLR-4, to modulate immune cell activation and tissue damage responses [35]. Their recognition is primarily mediated by pattern recognition receptors (PRRs) such as TLRs [36,37].

Upon recognizing DAMPs, TLRs initiate signaling cascades that result in the production of inflammatory cytokines, type I interferons, and other mediators, facilitating an effective immune response against damage [36]. These processes underscore the sophisticated mechanisms by which the immune system distinguishes between self and non-self as well as between different types of cell death to maintain homeostasis and defend against various pathological conditions [37].

#### 1.4.3. Adaptor Proteins and Signaling Pathways and Functional Consequences of TLR Activation

TLR signaling pathways are central to initiating both innate and adaptive immune responses. These pathways involve the recruitment of specific adaptor proteins, leading to the activation of downstream transcription factors that regulate immune and inflammatory responses. Each TLR can initiate distinct signaling cascades through one or both of the main pathways: the MyD88-dependent pathway and the TRIF-dependent pathway. All TLRs, except for TLR3, primarily utilize the MyD88-dependent pathway. This pathway involves the adaptor protein MyD88 (myeloid differentiation primary response 88) and leads to the activation of NF-κB, a key transcription factor that drives the expression of genes involved in inflammation and immunity, such as TNF-α, IL-1, and IL-6. The rapid activation of NF-κB facilitates immediate inflammatory responses, which are crucial for the initial phase of pathogen defense [38].

In contrast, the TRIF-dependent pathway, which is exclusive to TLR3 and is also utilized by TLR4, is initiated from TLR4-containing endosomes and results in the activation of interferon regulatory factors (IRFs), particularly IRF3. This pathway leads to the production of type I interferons, like IFN-α and IFN-β, which are vital for antiviral defense. The activation of IRFs contributes to a sustained immune response, suitable for combating viral infections and for modulating the immune system over a longer period [39]. The distinct and non-redundant roles of these pathways are critical for tailored immune responses, enabling the immune system to effectively handle a variety of pathogens. Understanding these pathways’ unique and overlapping functions is crucial for developing therapeutic interventions for immune-related diseases.

Upon recognition of PAMPs or DAMPs, TLRs trigger these signaling pathways through the recruitment of specific adaptor proteins, namely, MyD88, TIRAP (MAL), TRIF, TRAM, and SARM, which are crucial for downstream signaling. This cascade results in the activation of various transcription factors, including NF-κB, interferon regulatory factors (IRFs), and activator protein 1 (AP-1), leading to the expression of genes that drive the production of immune response molecules such as pro-inflammatory cytokines (e.g., TNF-α, IL-1β, IL-6), type 1 interferons (IFN-α and IFN-β), chemokines (CXCL8, CXCL10), and antimicrobial peptides [21,40]. The intricate network of TLR signaling pathways is illustrated in Figure 1. The consequences of these signaling events are significant, not only in combating infections but also in influencing host behavior and physiology. Cytokines produced in response to TLR activation can induce systemic effects such as weakness, lethargy, fatigue, and anorexia. These cytokines also orchestrate metabolic shifts towards catabolism, affecting muscle protein synthesis, inducing muscle wasting, and altering lipid metabolism by promoting lipolysis, upregulating fatty acid synthesis, increasing hepatic triglyceride production, and elevating serum triglyceride levels. Such extensive physiological changes underscore the pivotal role of TLRs in managing not just the immune response but also the metabolic adaptations during infections [21]. 

Furthermore, TLRs bridge the gap between innate and adaptive immunity, highlighting their importance in a comprehensive immune defense strategy. The complexity and regulation of these pathways ensure that the immune responses are appropriate and proportional to the threat, preventing overactivation and potential autoimmunity [41].

### 1.5. Role of TLRs in Infection 

#### 1.5.1. Recognition of Microbial Pathogens

TLRs play a crucial role in the immune system by recognizing PAMPs derived from microbes, thus initiating key signaling pathways that lead to immune responses. This recognition involves the specific binding of PAMPs, such as lipopolysaccharides from bacteria and viral RNA, to corresponding TLRs, which then activate transcription factors like NF-κB and IRFs. These factors are vital for the production of pro-inflammatory and antiviral cytokines and chemokines, essential for orchestrating the first line of defense in the immune system. The maturation of dendritic cells (DCs), triggered by these signaling events, is critical for T cell activation and the effective clearance of pathogens. These interactions highlight the indispensable role of TLRs in linking innate and adaptive immunity, underscoring their importance in immune surveillance. Enhanced understanding of TLR pathways, particularly their role in detecting microbial components, can significantly influence therapeutic strategies aimed at modulating immune responses for better disease management. The interaction between mature DCs and T cells is a fundamental aspect of the immune response that facilitates the activation and differentiation of T cells, which are critical for clearing pathogens [41,42,43,44].

TLR signaling pathways directly regulate effector and regulatory T cells (Tregs), highlighting the diverse roles of TLRs in both the innate and adaptive arms of the immune system. These pathways are involved in various diseases, including infectious diseases, autoimmune diseases, and cancer, highlighting the importance of TLRs in maintaining immune homeostasis and pathogenesis [45]. The interaction between TLRs and T cells is not unidirectional; T cell responses can be modulated by TLR signaling in a complex interplay that involves not only direct effects on T cells but also indirect effects mediated through antigen-presenting cells such as DCs. TLR-induced DC activation requires intrinsic complement pathways, further underscoring the complexity of TLR signaling and its impact on the immune response [41].

This intricate network of interactions among TLRs, DCs, and T cells underscores the essential role of TLRs in orchestrating a coherent immune response, linking innate immunity with the adaptive immune system to effectively respond to infections and other challenges.

#### 1.5.2. Initiation of Innate Immune Responses

TLRs play a pivotal role in the first line of immune system defense, distinguishing between self and non-self through the recognition of PAMPs derived from microbial entities and damaged cell components. Among these, TLRs located on the cell surface, such as TLR1, TLR2, TLR4, and TLR5, and those found in intracellular compartments, such as TLR3, TLR7, TLR8, and TLR9, provide a comprehensive system that allows for nuanced and highly specific responses to a wide array of microbial entities.

TLR2 and TLR4 have been recognized for their roles in identifying the components of the bacterial cell wall and distinguishing between Gram-positive and Gram-negative bacteria. TLR4 is known for its response to LPS, a major component of the outer membrane of Gram-negative bacteria, which initiates a signaling cascade that leads to the production of pro-inflammatory cytokines. Conversely, TLR2 is implicated in the recognition of peptidoglycans, lipoteichoic acids, and other components, predominantly in Gram-positive bacteria [46,47].

The interaction of TLR2 with its ligands plays a crucial role in the immune response to various pathogens, including Helicobacter pylori, a Gram-negative bacterium associated with gastric ulcers and cancer. This demonstrates the versatility and specificity of TLR2 in recognizing and responding to different microbial patterns [48]. Research has expanded our understanding of TLR2’s role in immune responses across different species, highlighting its evolutionary importance in host defense mechanisms [49].

The signaling pathways activated by TLR engagement are complex and highly regulated and involve a range of adaptor proteins and downstream signaling molecules. These pathways ensure that the immune response is tailored appropriately to the nature of the microbial threat. TLR signaling specificity is often determined by TIR domain-containing adaptors, which play a crucial role in translating the recognition of microbial components into appropriate cellular responses [50].

In conclusion, TLRs, particularly TLR2 and TLR4, are critical for the immune system to distinguish between different types of pathogens and initiate specific responses. Their ability to recognize a wide range of PAMPs underlies the effectiveness of the innate immune response as the first line of defense against microbial infections.

#### 1.5.3. Connection between Innate and Adaptive Immunity 

Intracellular TLRs, specifically TLR3, TLR7, TLR8, and TLR9, are located in the endosomal compartments of cells and play pivotal roles in the recognition of foreign nucleic acids by the innate immune system. This process is essential for initiating a robust antiviral response via the production of type I interferons and inflammatory cytokines, as illustrated in Figure 1. These TLRs serve as crucial sentinels in detecting pathogen invasions, enabling the body to mount an immediate defense against infections and to effectively bridge innate and adaptive immune responses [51,52,53].

TLR3 is unique in its ability to recognize double-stranded RNA, a molecular pattern associated with viral infection. TLR3 activation leads to the recruitment of the adaptor molecule TRIF, initiating a signaling cascade that ultimately results in the production of type I interferons and pro-inflammatory cytokines, thereby establishing an antiviral state within the host [53]. The crosstalk between TLRs and other receptor families, such as RIG-I-like receptors, further emphasizes the integrated nature of the innate immune response, allowing for the coordinated activation of antiviral defenses [52].

Moreover, conventional dendritic cells (cDCs) expressing TLR2, TLR4, TLR5, and TLR6 play significant roles in connecting innate and adaptive immunity. These TLRs, located on the cell surface, are crucial for the recognition of microbial components such as lipoproteins and lipopolysaccharides, leading to the activation of signaling pathways that promote the maturation of dendritic cells. This maturation is vital for effective antigen presentation to T cells, facilitating the activation and differentiation of T cells into effector cells that are central to adaptive immunity. Specifically, TLR4-mediated signaling through the NF-kB pathway is essential for the production of key cytokines and the upregulation of co-stimulatory molecules, which are critical for initiating a robust immune response [54]. Additionally, aqueous extracts of Artemisia rupestris L. have been shown to activate TLR4/TLR2, enhancing the phenotypic maturation of dendritic cells and promoting their ability to stimulate T-cell proliferation, further underscoring the pivotal role of these TLRs in the immune-modulating activities of dendritic cells [55].

Engagement of these intracellular TLRs not only triggers immediate innate immune responses but also influences the adaptive immune system. Activation of dendritic cells by TLRs results in the maturation of these antigen-presenting cells, enhancing their ability to present antigens to T cells and facilitating the development of a tailored adaptive immune response. The interplay between innate and adaptive immunity underscores the critical role of TLRs in maintaining immune surveillance and defense mechanisms against pathogens [10].

TLRs are instrumental in the development of immunotherapeutic strategies, including the formulation of vaccine adjuvants that target these receptors to enhance vaccine efficacy. By activating TLRs, these adjuvants can augment the immunogenicity of vaccines and promote stronger and more durable immune responses against cancers and infectious diseases [10].

In conclusion, the integration of innate and adaptive immune responses mediated by intracellular TLRs highlights the sophistication of immune system defense mechanisms. Through the recognition of pathogen-derived nucleic acids and the subsequent activation of immune responses, TLRs exemplify the intricate balance and communication between the body’s first line of defense and its more specific adaptive immune responses [56].

### 1.6. TLRs in Inflammation

#### 1.6.1. TLR-Mediated Inflammatory Pathways

TLRs are crucial for orchestrating the inflammatory pathways that are vital for immune responses to pathogens. Recent studies have underscored the crucial role of TLR4 in mediating severe inflammatory responses in conditions such as COVID-19, where inflammatory signaling pathways are markedly upregulated, contributing to disease severity [57]. Furthermore, innovative therapeutic strategies involving peptides derived from TIRAP, which act as decoys for TLR4, have shown promise in reducing inflammatory and autoimmune symptoms in murine models, offering potential for clinical applications [58]. They recognize distinct PAMPs, which initiate signaling cascades that activate transcription factors such as NF-κB and AP-1. These transcription factors drive the production of pro-inflammatory cytokines, chemokines, and other mediators involved in the inflammatory response. TIRAP (MAL), a key adaptor in TLR signaling, plays a critical role in recruiting and activating MyD88, which is essential for signaling through TLR2 and TLR4, thus promoting inflammatory cytokine production via NF-κB and AP-1 activation [59,60].

Intracellular signaling in the context of TLR activation involves a complex network of adaptors and kinases. Among these, the interaction between TIRAP and TRAF6 is critical for TLR2- and TLR4-mediated pro-inflammatory responses, illustrating the importance of protein–protein interactions in the specificity and amplification of TLR signaling pathways [61]. MyD88, a central adaptor molecule in innate immune signaling, plays a pivotal role in the downstream signaling of most TLRs as well as the IL-1 receptor family, connecting receptor engagement with the activation of IRAK family kinases. The organization of MyD88 domains, including the death domain, intermediate domain, and Toll-interleukin-1 receptor domain (TIR), is essential for its function in signal transduction [62].

Furthermore, TIRAP’s preferential localization to the cytoplasmic membrane through a PIP2-binding domain highlights the spatial regulation of TLR signaling, emphasizing the importance of membrane localization in the initiation of specific signal transduction pathways. Myristoylated TRAM, another adaptor molecule, localizes to endosomes and triggers type I interferon production via TRIF, delineating a clear distinction in the signaling roles and pathways mediated by different TLR adaptors [63].

This sophisticated regulatory mechanism ensures that the immune response is appropriately modulated, thereby preventing excessive inflammation that can lead to tissue damage or autoimmunity. The intricate interplay between these molecules exemplifies the complexity of the innate immune response and provides multiple targets for therapeutic intervention in diseases characterized by dysregulated inflammation or immune responses.

#### 1.6.2. Impact on Autoimmune Diseases

TLRs play complex roles in the development of autoimmune diseases, demonstrating distinct and varying implications across different conditions. TLR2, TLR4, TLR7, and TLR9 recognize PAMPs from infectious agents and DAMPs from tissue damage, contributing uniquely to the pathogenesis of specific autoimmune diseases. For example, TLR7 is prominently involved in the immune response mechanisms of systemic lupus erythematosus (SLE) by promoting the production of type I interferons, which are crucial in the disease’s progression. Similarly, TLR9 has been linked to the stimulation of autoreactive B cells in SLE, illustrating a direct influence on autoimmune pathology [64].

Alterations in TLR signaling, such as those resulting from gene polymorphisms, can significantly affect the immune balance, leading to diverse clinical outcomes in autoimmune conditions. Dysfunctions in TLR2, for instance, often result in decreased autoimmune inflammation, while overexpression might exacerbate it, as seen in diseases like rheumatoid arthritis and multiple sclerosis [23]. TLR signaling extends beyond innate cells to adaptive immune cells such as T and B lymphocytes, which utilize TLRs to modulate responses and affect autoimmune disease progression.

Recent research has further demonstrated that inhibitors like TAC5 can selectively target endosomal TLRs (TLR3, TLR7, TLR8, and TLR9) to modulate immune responses and potentially treat autoimmune diseases like psoriasis and systemic lupus erythematosus. For instance, a novel selective Toll-like receptor 7/8 inhibitor, known as M5049, has shown potential in treating autoimmunity by blocking multiple TLR7/8 RNA ligands with significant efficacy in murine lupus models [65]. Additionally, modulation of TLR signaling by small-molecular inhibitors such as those targeting TLR7, TLR8, and TLR9 has shown promise in ameliorating symptoms of autoimmune disorders, as illustrated by the discovery of potent and orally bioavailable small-molecular antagonists of TLR7/8/9, which have demonstrated efficacy from oral dosing in the preclinical models of autoimmune diseases [66]. The complex interaction between various TLRs, including their non-canonical roles in immune system modulation, continues to be a crucial area of study with significant therapeutic potential.

### 1.7. Advancements in TLR Research

#### Recent Discoveries and Technological Advancements

The recognition of PAMPs and DAMPs by TLRs initiates signaling cascades that significantly affect the immune balance, leading to either the mitigation or exacerbation of autoimmune inflammation. TLRs, such as TLR2, TLR4, TLR7, and TLR9, are involved in this process, and their dysfunction or overexpression in immune cells plays a pivotal role in the pathogenesis of autoimmune diseases. 

TLR signaling pathways are complex and extend beyond innate immune cells to adaptive immune cells, such as T and B lymphocytes, affecting autoimmune disease progression. For instance, alterations in TLR signaling can lead to the inappropriate triggering of TLR pathways by exogenous or endogenous ligands, causing the initiation and/or perpetuation of autoimmune reactions and tissue damage. This underlines the importance of TLRs in both regulatory and inflammatory pathways within the context of autoimmune disease pathogenesis [63,64,67]. Recent studies have further highlighted novel technological advancements in TLR research that significantly enhance our understanding and potential therapeutic applications. Notably, the development of TLR therapeutics has shown promise in the treatment of chronic viral infections, with TLR7 agonists currently in clinical trials for chronic HBV infection and a TLR9 ligand used as an adjuvant in a superior HBV vaccine [68]. Additionally, the creation of a highly sensitive TLR reporter platform represents a significant technological advancement. This platform employs a sensitive NF-κB::eGFP reporter cell line for the detection of TLR ligands with high specificity, which is crucial for ensuring the purity of therapeutic compounds and understanding TLR ligand interactions [69]. These technological innovations not only deepen our understanding of TLR roles in immune modulation but also open new avenues for the development of targeted therapies for autoimmune and infectious diseases. Given their crucial roles in infectious and non-infectious disease processes, TLRs and their signaling pathways have emerged as attractive targets for therapeutic interventions. Strategies aimed at modulating TLR signaling could offer new avenues for treating autoimmune diseases by restoring immune balance and preventing tissue damage [70,71].

Recent advancements include the discovery of small-molecular antagonists, such as 7f, which exhibits strong on-target potency and selectivity against TLR7 and TLR9, demonstrating efficacy in the preclinical models of autoimmune diseases by oral administration [66]. Furthermore, the identification of the novel small-molecular TLR-inhibitor, TAC5, has shown to significantly reduce inflammation and alleviate symptoms in autoimmune disease models, indicating a potential therapeutic application for diseases such as psoriasis and SLE [13]. These insights into TLR signaling pathways in the context of autoimmune diseases highlight the need for further research to explore therapeutic options that target these pathways. Such approaches may provide novel treatments for autoimmune conditions by leveraging our understanding of TLR-mediated immune responses.

### 1.8. TLRs as Therapeutic Targets

#### 1.8.1. Potential in Drug Development

TLRs have significant potential in drug development, particularly for their roles in immunotherapy, cancer treatment, and inflammation control. Recent advances in immunotherapy have highlighted the efficacy of TLR agonists, such as Poly-ICLC, in enhancing anticancer immune responses. These agents function by stimulating TLR3, leading to enhanced expression of inflammatory genes and facilitating the infiltration of T cells into tumors, thus demonstrating significant potential in cancer treatment [72]. Their capacity to recognize PAMPs and DAMPs makes them attractive targets for modulating immune responses as illustrated in Figure 1.

Poly-ICLC, a well-studied TLR3 agonist, has been highlighted for its immunostimulatory effects, which can be leveraged to induce anticancer immune responses. Poly ICLC functions by stimulating TLR3 and cytosolic MDA5, leading to the activation of various pattern recognition receptors. This activation results in the enhanced expression of inflammatory genes and infiltration of T cells into tumors, illustrating its multifaceted role in cancer immunotherapy. The uniqueness of poly-ICLC lies in its ability to engage different receptors, thereby eliciting a broad immunomodulatory response that can overcome barriers to effective cancer treatment [72].

Further exploration of TLR3 agonists, such as RGC100, ARNAX, and poly-IC, revealed distinct mechanisms and shared characteristics. These agonists mimic double-stranded RNA and trigger pathways that contribute to cancer regression. ARNAX activates TLR3 but not the cytoplasmic receptor MDA5/RIG-I, inducing the Toll-like receptor 3–Toll-interleukin-1 receptor domain-containing adaptor molecule 1–interferon regulatory factor 3–interferon-α/β receptor (TLR3-TICAM-1-IRF3-IFNAR) signaling axis, which is crucial in dendritic cells for cytotoxic T lymphocyte induction. Such specificity and ability to enhance anti-tumor responses, especially in conjunction with PD-1/PD-L1 blockade, underscore the therapeutic promise of TLR3 agonists in cancer treatment [73].

Additionally, the development of chemically defined TLR3 agonists, such as TL-532, further demonstrates ongoing innovation in this field. TL-532, which is composed of poly(I:C) and poly(A:U) blocks, has shown potential because of its favorable pharmacokinetic, pharmacodynamic, and toxicological properties. In preclinical models, TL-532 reduces tumor growth and restores the efficacy of immunogenic chemotherapy, making it a promising candidate for anticancer immunotherapy [74].

Moreover, several TLR9 ligands have been effectively utilized as vaccine adjuvants and in combination with other anti-tumor drugs for cancer therapy. The dual-adjuvant effect of liposomes loaded with TLR9 and other immune stimulants has shown significant potential in regressing tumor development and enhancing Th1 immune responses in clinical settings [75]. The strategic use of these TLR9 ligands underscores their adaptability and efficacy in a broad range of therapeutic applications, providing a significant enhancement to both innate and adaptive immunity [10].

Furthermore, recent advancements in the development of dual TLR7/9 antagonists have opened new avenues for therapeutic applications in autoimmune diseases due to their selective inhibition properties [76]. Additionally, sugar-conjugated TLR7 ligands have shown increased immunostimulatory activity and have potential applicability in vaccine adjuvant and cancer therapy contexts [77]. As summarized in Table 3, TLRs play crucial roles in infection and inflammation. These examples highlight the evolving landscape of TLR-targeted therapies and their broadening applicability across different medical fields.

Overall, advancements in TLR-targeting strategies, including the development and clinical trials of specific ligands and biologics, illustrate the dynamic and promising fields of immunotherapy. Through modulation of TLR activity, these approaches offer a versatile toolkit for enhancing immune responses against cancers, further enriching the landscape of therapeutic options available for cancer treatment. For a detailed list of these clinical trials, please refer to Table 4.

#### 1.8.2. Current Clinical Trials and Future Prospects

Recent clinical trials have shown the potential of TLR agonists and antagonists in treating a broad spectrum of diseases by modulating the immune system. These developments have led to significant progress in leveraging the immunomodulatory capabilities of TLRs for therapeutic interventions.

One study investigated the safety and pharmacodynamics of the TLR7 agonist GSK2245035 in patients with respiratory allergies. This study aimed to explore the potential of TLR7 agonists in altering immune responses in the upper airways, which may contribute to the reduction in allergic reactivity [78]. Similarly, another study evaluated MEDI9197, a TLR7/8 agonist, when administered alone or in combination with durvalumab and/or palliative radiation in subjects with solid tumors, highlighting the exploration of TLR agonists in cancer therapy [73,79]. Moreover, the study on CYT003-QbG10 for treating allergic bronchial asthma is another example of TLR-targeted therapy aiming to improve asthma symptoms by potentially modulating the immune system [80]. Additionally, TLR agonists are also being utilized as vaccine adjuvants targeting cancer and infectious diseases, demonstrating their significant potential in enhancing the immune response to purified antigens, thus broadening protection against highly variable pathogens [10].

Moreover, recent trials have explored the use of various TLR antagonists and agonists. Among the promising TLR7 antagonists in development are compounds from the imidazo[1,2-a]pyrazine, imidazo[1,5-a]quinoxaline, and pyrazolo[1,5-a]quinoxaline series, which have been shown to be potent and selective without any TLR7/8 agonistic activity, offering potential therapeutic applications in autoimmune and infectious diseases [81].

Additional ongoing studies include the examination of cholesterolized liposome formulations of TLR7 agonists, which have shown potential in reducing tumor growth through effective delivery and activation of immune responses within the lymph nodes [82]. These clinical trials represent only a fraction of the ongoing research efforts to harness TLRs’ immunomodulatory potential across various diseases, including allergies, asthma, cancer, and chronic infections. The therapeutic applicability of TLR agonists and antagonists across a wide range of conditions underscores their promise of TLRs as targets for future medical interventions, potentially offering new treatments for chronic inflammation, autoimmunity, and other conditions. The results of these trials could pave the way for novel therapeutic strategies that leverage the intricate mechanisms of the immune system to fight diseases.

These clinical trials represent only a fraction of the ongoing research efforts to harness TLRs’ immunomodulatory potential across various diseases, including allergies, asthma, cancer, and chronic infections. The therapeutic applicability of TLR agonists and antagonists across a wide range of conditions underscores their promise of TLRs as targets for future medical interventions, potentially offering new treatments for chronic inflammation, autoimmunity, and other conditions. The results of these trials could pave the way for novel therapeutic strategies that leverage the intricate mechanisms of the immune system to fight diseases.

### 1.9. Challenges and Controversies 

#### 1.9.1. Limitations in Current Research

TLRs are critical components of the innate immune system that recognize a broad spectrum of molecular patterns to initiate an immune response. Each of the ten functional TLRs (TLR1–10) in humans plays a pivotal role in detecting various dangers and pathogen-associated molecular patterns, including DAMPs, MAMPs, PAMPs, and XAMPs. These receptors share a common domain organization that includes an N-terminal domain with LRRs, a single TM, and a C-terminal cytoplasmic TIR domain [1,64].

Recent studies have elucidated the significance and multi-functionality of TLRs in innate immunity. TLR9 and cyclic GMP-AMP synthase-stimulator of interferon genes (cGAS-STING) have been recognized for their roles in DNA sensing and the initiation of innate immune responses, highlighting the complexity and adaptability of TLR-mediated signaling in response to various stimuli [83]. Furthermore, the importance of TLRs in viral infections has been underscored by the finding that TLR3, TLR7, and TLR8, along with cytoplasmic RIG-I-like receptors, facilitate the initiation of innate immune responses by recognizing viral nucleic acids [84].

Moreover, research on TLR signaling has revealed its impact on the progression of diseases, such as herpes simplex virus 1 (HSV-1) infection, where TLRs and their signaling pathways in the trigeminal ganglia are critical for controlling the progression of the virus to encephalitis [85]. These findings demonstrate the essential role of TLRs in antiviral defense and their potential as therapeutic targets.

Activation of TLR pathways, along with other innate immune pathways, has been exploited for therapeutic purposes, such as in the development of adjuvant-free subunit vaccines. One study demonstrated the use of a polymer that activates both TLR and cGAS pathways, inducing a strong humoral immune response that is pivotal for effective vaccination strategies [86].

In summary, TLRs are indispensable for recognizing a wide array of molecular patterns that initiate critical defense responses in the innate immune system. Ongoing research and discoveries in the field of TLR signaling pathways continue to reveal the intricate mechanisms through which the innate immune system senses and responds to microbial and non-microbial threats, thereby reinforcing the potential for TLR-targeted therapeutic interventions.

#### 1.9.2. Debated Roles of TLRs in Various Diseases

The complex roles of Toll-like receptors (TLRs) in various diseases have been a subject of ongoing debate, as evidenced by their contradictory effects on disease progression and therapeutic outcomes. In autoimmune diseases, particularly in systemic lupus erythematosus (SLE), TLRs may worsen the condition by promoting inflammatory responses. Conversely, in cases of infection-induced autoimmunity, TLRs might serve a protective role by helping to eliminate pathogens. This ambiguity is highlighted in neuroimmune diseases, where TLRs can both exacerbate and mitigate disease symptoms depending on the context [87]. In the context of cancer, TLRs exhibit a dual functionality; their activation might suppress tumor growth by enhancing immune surveillance, yet it could also facilitate tumor progression by creating an immune-tolerant environment [88,89]. The role of TLRs in neurodegenerative diseases remains particularly complex and is still under investigation. Some studies suggest that TLR activation could be neuroprotective by aiding in the clearance of pathogenic proteins, while others argue that it may lead to neuroinflammation and neuronal damage [90]. These examples underscore the need for a more refined understanding of TLRs’ roles, which is crucial for advancing both research and clinical interventions in these areas.

### 1.10. Future Directions

#### 1.10.1. Emerging Research Areas

Future research is focused on their therapeutic potential, particularly in immunotherapies. Studies have focused on TLR agonists and inhibitors as dual-purpose agents in cancer therapy, infectious diseases, and autoimmune conditions, aiming to refine immune responses in various diseases with a keen interest in personalized medicine [91]. Research is progressing, with clinical trials and computational models to enhance the efficacy and specificity of TLR-based therapies.

TLR agonists have shown promise for enhancing the immune response to purified antigens in vaccines against life-threatening and complex diseases such as cancer, AIDS, and malaria [10]. In head and neck squamous cell carcinoma (HNSCC), a combination of intratumoral injections of TLR7 and TLR9 agonists with PD-1 blockade suppresses tumor growth and promotes systemic adaptive immunity. This approach activates tumor-associated macrophages, induces tumor-specific immune responses, and inhibits metastasis in HNSCC models [91].

In oncological treatments, TLR agonists elevate PD-L1 levels within the tumor microenvironment, potentially augmenting the efficacy of checkpoint inhibitor therapies [92]. Employing TLR agonists in tandem with checkpoint inhibitors can effectively mitigate head and neck cancer, demonstrating the efficacy of localized TLR agonist application alongside anti-PD-1 therapy in fostering systemic adaptive immune responses [91].

For infectious diseases, TLR agonists have been formulated as licensed vaccines for their adjuvant activity, and other TLR agonists have been developed for this purpose [93]. Monophosphoryl lipid A and CpG-1018 have been used as adjuvants in vaccines against viral infections, including hepatitis B virus, hepatitis C virus, human immunodeficiency virus, SARS-CoV-2, influenza virus, and flaviviruses [94].

In autoimmune conditions, TLR agonists have been linked to the perpetuation of inflammation in chronic inflammatory diseases due to the permanent exposure of the immune system to TLR-specific stimuli [95]. However, recent studies have demonstrated that MSCs are activated by TLR ligands, leading to the modulation of differentiation, migration, proliferation, survival, and immunosuppression. This activation has not been reported to modulate the “immunoprivileged” phenotype of MSCs, which is of particular relevance for the use of allogeneic MSC-based therapies [95].

In summary, future TLR research should focus on their therapeutic potential in immunotherapies, particularly in cancer therapy, infectious diseases, and autoimmune conditions. Studies are progressing with clinical trials and computational models to enhance the efficacy and specificity of TLR-based therapies, with a keen interest in personalized medicine.

#### 1.10.2. Potential Breakthroughs

TLRs are increasingly being recognized as critical players in the development of innovative therapeutic strategies, particularly in oncology and autoimmune disease management. As key components of the immune system, TLRs are involved in the recognition of pathogen-associated molecular patterns (PAMPs) and damage-associated molecular patterns (DAMPs), which are crucial in initiating immune responses [96,97]. Their ability to modulate both innate and adaptive immunity makes them valuable targets for new cancer therapies aimed at harnessing the body’s defense mechanisms to combat malignancies [98].

In the context of cancer, TLRs can influence tumor progression and the ability of the immune system to recognize and destroy cancer cells. The dual nature of TLRs in cancer, in which they can either promote or inhibit tumor growth, underscores the complexity of their roles in oncogenesis and the need for precise therapeutic targeting [96,97]. For example, TLR agonists have shown promise in enhancing the efficacy of cancer vaccines in combination with checkpoint inhibitors to stimulate anti-tumor immunity [96,98].

Dysregulation of TLR signaling can lead to persistent inflammation and tissue damage in autoimmune diseases. TLRs are being explored as potential therapeutic targets to dampen excessive immune responses without compromising the body’s ability to fight infections [23]. Understanding the genetic polymorphisms within TLRs that contribute to autoimmune susceptibility emphasizes the importance of personalized medical approaches for treating these conditions [23].

Computational modeling has revolutionized TLR research by providing sophisticated tools for predicting the outcomes of TLR signaling pathways and the impact of potential therapeutics [99]. These models can simulate the dynamic interactions within the immune system, allowing researchers to explore the effects of TLR modulation in silico before proceeding with in vivo studies. This approach can streamline drug development, reduce the risk of adverse effects, and accelerate the translation of research findings into clinical applications [99].

In summary, the integration of TLR biology with computational innovations is expanding therapeutic possibilities for a range of medical conditions, from infectious diseases to cancer and autoimmune disorders. Researchers are paving the way for more targeted and effective treatments that can transform patient care by leveraging the ability of TLRs to regulate immune responses.

## 2. Conclusions

We have summarized the critical roles that TLRs play within the innate immune system. These receptors are instrumental in detecting microbial patterns and serve as the first line of defense against pathogens. Our discussion extends the molecular intricacies of TLRs, including their structure and signaling pathways, which are fundamental to their function in immune surveillance and disease pathogenesis.

The implications of TLRs in a spectrum of diseases are the focus of our review. In autoimmune disorders, aberrant TLR signaling can lead to inappropriate immune responses against tissues, whereas in chronic inflammatory diseases, persistent TLR activation contributes to ongoing tissue damage. In oncology, TLRs can have a dual role, either supporting tumor progression through inflammatory pathways or enhancing anti-tumor immunity, depending on the context and specific TLRs involved.

The therapeutic potential of TLRs is vast, particularly highlighted by the development of controlled TLR agonists and antagonists that modulate immune responses. These agents are being actively explored for their capacity to either boost the immune response in cases such as cancer and infectious diseases or dampen it in autoimmune and inflammatory conditions. Specifically, TLR agonists such as those targeting TLR7/8 have shown promise in enhancing the immunogenicity of vaccines against severe diseases like cancer, AIDS, and malaria by linking innate with adaptive immune responses [10]. Additionally, novel therapeutic applications are being developed for TLR10 due to its unique anti-inflammatory properties, suggesting its potential use in therapeutics targeting inflammatory and autoimmune diseases [100].

As we look to the future, the ongoing research into TLRs is set to revolutionize the field of medicine. The integration of TLR-based therapies with other modalities, such as checkpoint inhibitors in cancer or biologics in autoimmune diseases, is likely to enhance treatment efficacy and patient outcomes. Moreover, continuous advancements in computational biology and genomics will enable a deeper understanding of TLR signaling and its implications in health and disease, paving the way for more personalized and targeted therapeutic interventions.

In conclusion, TLRs represent a promising frontier in medical research with the potential to significantly impact the development of novel therapeutic strategies. Their roles in the immune system and disease pathogenesis offer a rich landscape for scientific exploration, with the promise of yielding new insights that could transform the practice of medicine.

## Figures and Tables

**Figure 1 ijms-25-05037-f001:**
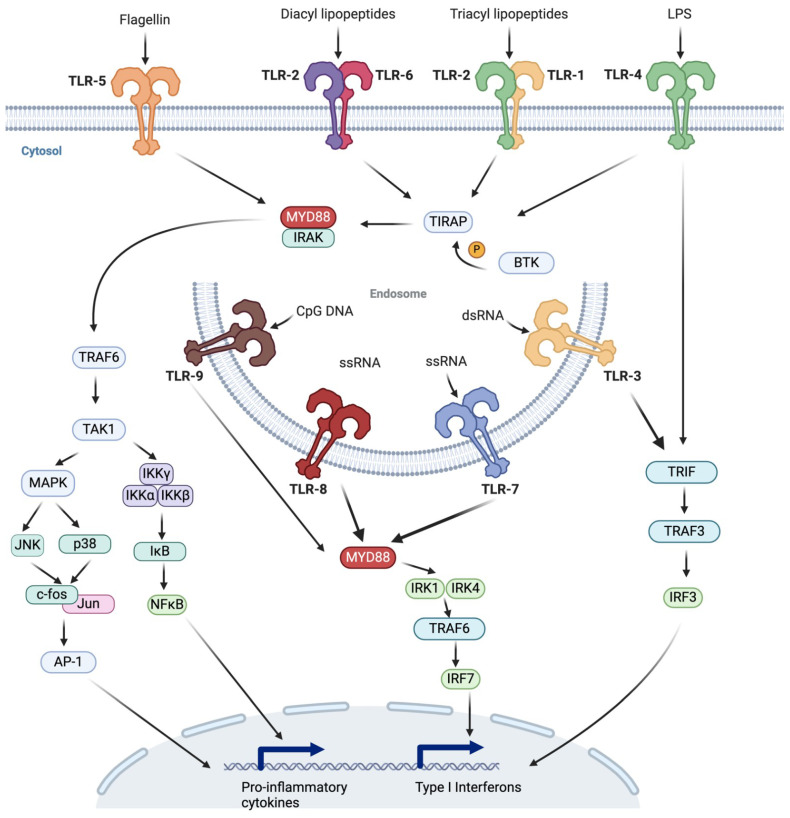
Overview of Toll-like receptor (TLR) signaling pathways. This schematic depicts the cell-surface TLRs (TLR-1, TLR-2, TLR-4, TLR-5, and TLR-6) and their ligands (flagellin, diacyl lipopeptides, triacyl lipopeptides, and LPS). It also shows endosomal TLRs (TLR-3, TLR-7, TLR-8, and TLR-9) recognizing CpG DNA and RNA molecules. The diagram illustrates the subsequent signaling cascades involving adaptor proteins such as MYD88, TRIF, and TRAF6, leading to the activation of NF-κB and IRFs and resulting in the expression of pro-inflammatory cytokines and type I interferons.

**Table 1 ijms-25-05037-t001:** Therapeutic potential of TLRs.

Approach	Examples	Diseases	Reference
TLR Agonists	Poly-ICLC, MEDI9197, QbG10, Resiquimod, Imiquimod	Cancer, infectious diseases, allergies, HPV	[10,11,12]
TLR Antagonists	TAC5, MHV370	Autoimmune diseases (psoriasis, SLE), inflammatory conditions	[13,14]
Vaccine Adjuvants	Monophosphoryl Lipid A (MPL), CpG-1018, Resiquimod, Imiquimod	Infectious diseases (HBV, HCV, HIV, SARS-CoV-2, influenza, HPV)	[10,11,12]

**Table 2 ijms-25-05037-t002:** Overview of Toll-like receptors (TLRs).

TLR	Primary Location	Principal Ligand	Signaling Pathway	Functions and Application Examples
TLR1/2	Cell Surface	Triacylated lipopeptides (Pam3CSK4)	MyD88-dependent	Involved in immune response activation, potential applications in cancer and infectious disease treatments [1]
TLR2/6	Cell Surface	Lipopeptides	MyD88-dependent	Regulates inflammation and immune responses, potential treatments for allergic and autoimmune diseases [30]
TLR3	Endosomes	Double-stranded RNA (dsRNA)	TRIF-dependent	Antiviral immune responses, applications as vaccine adjuvants and in tumor treatments [31]
TLR4	Cell Surface	Lipopolysaccharide (LPS)	MyD88-dependent and TRIF-dependent	Regulates inflammatory responses, potential treatments for infectious diseases and cancer [32]
TLR7	Endosomes	Single-stranded RNA (ssRNA)	MyD88-dependent	Antiviral responses and regulation of autoimmune diseases such as SLE [13]
TLR8	Endosomes	Single-stranded RNA (ssRNA)	MyD88-dependent	Antiviral and anti-tumor therapeutic applications [13]
TLR9	Endosomes	CpG DNA	MyD88-dependent	Applications in antiviral vaccine adjuvants, treatments for infectious diseases and tumors [30]
TLR10	Cell Surface	Double-stranded RNA (dsRNA) (presumed)	MyD88-dependent	Potential roles in antiviral responses and immune regulation, further research needed to clarify specific functions [33]

**Table 3 ijms-25-05037-t003:** TLRs in infection and inflammation.

Role	Mechanisms	Implications
Innate Immunity	Recognize PAMPs and DAMPs, initiate signaling cascades	First line of defense against pathogens
Adaptive Immunity	Activate dendritic cells, modulate T cell responses	Bridge innate and adaptive immunity
Inflammatory Diseases	Aberrant TLR signaling, persistent activation	Autoimmunity, chronic inflammation
Cancer	Dual role in tumor progression and anti-tumor immunity	Potential therapeutic targets

**Table 4 ijms-25-05037-t004:** Future directions in TLR research.

TLR	Molecule	Type	Clinical Trial Status	Indications	Reference
TLR3	Poly-ICLC	Agonist	In Use	Cancer Immunotherapy	[72]
TLR3	RGC100	Agonist	Research Phase	Cancer Treatment	[73]
TLR3	ARNAX	Agonist	Research Phase	Cancer Treatment	[73]
TLR3	poly-IC	Agonist	Research Phase	Cancer Treatment	[73]
TLR3	TL-532	Agonist	Preclinical	Cancer Immunotherapy	[74]
TLR9	Various Ligands	Agonist	Clinical Settings	Cancer Therapy, Vaccine Adjuvant	[75]
TLR7	GSK2245035	Agonist	Clinical Trial	Respiratory Allergies	[78]
TLR7/8	MEDI9197	Agonist	Clinical Trial	Solid Tumors	[79]
TLR9	CYT003-QbG10	Agonist	Clinical Trial	Allergic Bronchial Asthma	[80]
TLR7	Imidazo derivatives	Antagonist	Phase I	Autoimmune, Infectious Diseases	[81]
TLR7	Cholesterolized Liposomes	Agonist	Preclinical	Cancer Treatment	[82]

## Data Availability

Not applicable.

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
