# Peer review of "Unraveling the Complexities of Toll-like Receptors: From Molecular Mechanisms to Clinical Applications"

_ijms, 2024, doi:10.3390/ijms25095037_

Round 1

Reviewer 1 Report

Comments and Suggestions for Authors

In this submitted paper, the authors tried to summarize the significance of Toll-like receptors (TLRs) in the pathogenesis of diseases and immune surveillance. After introducing structure, signaling pathways, and involvement in numerous disorders, the molecular complexities of TLRs, comprising ligand specificity, signaling cascades, and the activation consequences are described. Their potential as therapeutic targets is highlighted by exploring TLRs' contribution to infectious diseases, autoimmunity, chronic inflammation, and cancer. They examine the latest developments in TLRs, such as the improvement of agonists and antagonists, their applications in immunotherapy and vaccine development. Additionally, the challenges and controversies surrounding TLR research are addressed and future directions including the integration of computational modeling and personalized medicine approaches are drawn. The manuscript can be accepted for publication in the International Journal of Molecular Sciences. However, some major issues underlined in the following comments need to be resolved before further consideration.

Major issues:

1.      The abbreviations used throughout the paper must be rechecked carefully since some of them don’t have any corresponding full terms making it hard for the reader to figure out what is the exact meaning of the statements.

2.      The whole paper suffers from the lack of attractive and informative illustrations. On the other hand, a good review paper must have numerous images and figures to give an overview to the readers about each section or subsection discussion of the review paper.

3. The discussions provided for most parts of the paper are concise and must be expanded by adding appropriate scientific literature and research works recently published.

Reviewer 2 Report

Comments and Suggestions for Authors

The manuscript entitled “Unraveling the Complexities of Toll-Like Receptors: From Molecular Mechanisms to Clinical Applications” summarizes the molecular intricacies of TLRs and their involvement in various diseases, as well as insights in developing therapeutic strategies for certain diseases based on TLR’s crucial roles in innate immune responses. This review, although language is fluent, is not enough depth of contents and lack of illustrated figures, and the whole manuscript is not well organized, lots of sections are repeated concepts. The authors need to reorganize the manuscript before considering publication in IJMS.

Some comments for the authors:

1, Page 1-2, Line 31-61, since TLRs and RLRs are only a part of PRRs, when summarizing an overview of innate immunity and PRRs, the TLRs, RLRs, CLRs, NLRs and cGAS-Sting should all be included, not just TLRs, RLRs and cGAS.

2, Page 2, Line 104-116, regarding to TLR family members, highly recommend a summary of how many TLRs there are and what are the main functions of each TLR, and their expression map as well. For example, the authors did not mention TLR11, TLR12, and TLR13.

3, Page 4, Line 145, as DAMPs, galectins are important and act as ligands of TLRs, but the authors fail to include this.

4, Page 5, Line 198, the context of this paragraph is weird, the subtitle is Recognition of Microbial Pathogens, however the authors discuss the role of TLRs in liking innate immunity with adaptive immunity such as DCs and Tregs, but not the functions and ways of TLR recognition of pathogens.

5, Page 7, Line 330, the authors post a subtitle of “Technological advancement”, but none of new TLR-related technologies is stated in this part. Instead, the authors repeat the contents discussed previously.

6, There are lots of repeated concepts in different sections, for example: the contents of “1.2 Brief introduction to TLRs” and “1.3.2 Differences between TLR Family Members”; the contents of “1.4.3 Adaptor Proteins and Signaling Pathways”, “1.4.4 Functional Consequences of TLR activation”, “1.5.2. Initiation of Innate Immune Responses” and “1.6. TLRs in Inflammation”; and some others.

7, Page 7, Line 326, what does the “^1^” mean here?

Comments on the Quality of English Language

Fine

Reviewer 3 Report

Comments and Suggestions for Authors

This manuscript that submitted by Chen et al introduced the important role of TLRs signaling in innate immune system defensing invasion of pathogens and they summarized the clinical application by using TLRs ligand against diseases. (1) I suggest that authors should draw cartoon or TLRs signal pathway to help audiences more easy to understand that each TLR can recognize different ligand to trigger the downstream signaling. (2) Overall, this manuscript had a lot of space to introduce basic research of TLRs and just a little space for clinical application by using TLR ligands against diseases. (3) I further found that authors seemed to like introduce more TLR3 and TLR7 ligands in clinical applications, however, several TLR9 ligands had been used as a vaccine adjuvant or in combination with other anti-tumor drugs for cancer therapy. (4) The authors should summarize or focus on specific disease to emphasize the important role by activation of TLR signaling. In current version of manuscript, it look like that TLR activation may be a good strategy against some diseases, but I just observe a little space about clinical application on several diseases by using TLRs ligands. I cannot catch the significant impact on treating disease by TLR activation.     

Comments on the Quality of English Language

The scientific writing is good and easy to read.

Reviewer 4 Report

Comments and Suggestions for Authors

This article summarized comprehensive knowledge about Toll-Like receptors and aimed to address the challenges and controversies surrounding TLR research and outline future directions. Toll-like receptors (TLRs) are the essential molecules in the innate immunity, and are the targets in the development of novel therapeutic strategies for a wide range of diseases. Therefore, the rational of this article is convincing. However, the majority of contents provided only basic knowledge about the receptors.

(1) From section 1.1 to section 1.8: The contents are basic, and some are at textbook levels. This section can be shortened and should include more advanced knowledge.

(2) Section 1.1. why the section described only TLR3, TLR7, TLR8, and TLR9? If the authors would like to discuss nucleic acid-sensing TLRs, it is better to focus on it.

(3) Sections 1.4.3 and 1.4.4: It is better to add figure(s) to visualize specific signal transduction in TLRs.

(4) Section 1.5.3, the title is “Connection between Innate and Adaptive Immunity”. This section, again, described only TLR3, TLR7, TLR8, and TLR9. However, dendritic cells expressing TLR3, TLR7, and TLR9 (pDC) are not specialized for antigen presentation. cDC expressing TLR2, TLR4, TLR5, and TLR6 would play more important roles to connect between innate immunity and adaptive immunity.

(5) Section 1.6.2: Please add recent references.

(6) Section 1.7: Please describe how TLR Research has been advanced in detail.

(7) Section 1.8: There must be more molecules in current ongoing clinical trials, and should be described in texts and listed in a Table.

(8) Why only a limited number of TLR7 ligands were described in this section?

(9) Section 1.9.2: The title of section is "Debated Roles of TLRs in Various Diseases". Please describe how the roles of TLRs in various diseases is debated more in detail.

Reviewer 5 Report

Comments and Suggestions for Authors

The review presented by the group of Wu and collaborators focuses on analyzing the complexity of TLRs, introducing new perspectives and possible applications of them.

The text of the review is correct, but it is very descriptive without providing new aspects that could increase the interest of readers.

Main points:

1. The authors should present more visual diagrams of the archetypes of these receptors, showing the common and specific domains, as well as the signaling mechanisms associated with them. These aspects are included in a large number of works, but here they could serve to add the more innovative aspects to which the authors allude.

2. If the nature of the TLRs analyzed is not precise, the comments remain in a very generic field, without being able to specify the interest in developing their agonists and/or antagonists. In this sense, the tables presented are very vague and significantly reduce the interest of the work, since they do not allow readers to determine the fields in which the action on TLRs has pharmacological or therapeutic relevance.

3. Perhaps one of the aspects of greatest interest to readers is the critical analysis of clinical trials since in this way the information referred to in the title of the review is provided. There are many clinical trials in which TLRs are used as targets and, therefore, it would be very important to group them rationally; for example, reflecting pathologies, combined therapies, etc.

4. An aspect not addressed that is important in the case of translating the results of experimental models to human pathology is the analysis of the polymorphisms described for these molecules. This aspect is important and could bring novelty to the approach that the authors want to give to the review.

Round 2

Reviewer 1 Report

Comments and Suggestions for Authors

Accept

Author Response

Thank you!

Reviewer 2 Report

Comments and Suggestions for Authors

addressed

Author Response

Thank you!

Reviewer 3 Report

Comments and Suggestions for Authors

The authors included my suggestions to revised their previous manuscript and let this manuscript become more better. It seem likely to reach the criterion to publish.

Comments on the Quality of English Language

The scientific writing is good and easy to understand.

Author Response

Thank you!

Reviewer 4 Report

Comments and Suggestions for Authors

The quality of this manuscript was improved. However,  this reviewer still concerns following parts,

(1) Introduction: What is the aim of this review article? It is important to describe the aim of the article in the introduction, but the author described only general information about TLRs in this section.

(2) Line 190-187: Please rephrase texts. This would mislead as if signal pathways to pro-inflammatory cytokines and type I IFN are similar. It is important to explain how the pathways are different and spedific in induction of these cytokine expression.

(3) Figure 1: The figure is too simple and is not scientific. For instance IKK complex cannot be indicated as one molecule. In addition, there are more molecules involved in the cadcades. The words of sinaling molecules are too small to see.

(4)Table 3: The table is poor. It is very unlikely that only two TLR ligands are potential vaccine adjuvants.

(5)section 1.6.2. Impact on Autoimmune Diseases. The content is too general. This section would mislead as if all autoimmune diseases are triggered by TLR in similar mechanism. However, each autoimmune disease has different pathological mechanism. It is important to describe several examples  of autoimmune diseases, in which TLRs are deepley involved, and explain it at cellular and molecular mechanisms in more detail.

Reviewer 5 Report

Comments and Suggestions for Authors

The authors improved the review as requested.

Author Response

Thank you!

Round 3

Reviewer 4 Report

Comments and Suggestions for Authors

The authors replied to the reviewer's comments well. However, the quality of figure 1 still needs to be improved. This reviewer suggests the authors to see many figures in other publications for better scientific drawing.

(1) Location of MYD88 is far away from TLR1,2,4 and 6. Is the proximity of MYD88 and TLRs correct? 

(2) Why JNK and p38 form a complex?

(3) Pro-inflammatory gene expression should be pro-inflammatory cytokines?

(4) A TRIF and TRAF3 cascade of TLR3 and TLR4 is missing.

(5) The arrow head of NF-kB should enter into the nucleus. 
